# PROS: Towards Compute-Efficient RLVR via Rollout Prefix Reuse

**Baizhou Huang & Xiaojun Wan**
Wangxuan Institute of Computer Technology, Peking University
{hbz19,wanxiaojun}@pku.edu.cn

## Abstract

Large reasoning models (LRMs) trained with *Reinforcement Learning with Verifiable Rewards* (RLVR) have achieved remarkable progress on complex reasoning tasks. However, RLVR heavily relies on on-policy rollout generation, whose cost grows rapidly with rollout length and model size, eventually becoming the training bottleneck. Our empirical analysis reveals that independent rollouts for the same query often share similar early steps, indicating substantial redundancy. To address this, we propose PROS (**P**refix **R**euse for **O**n-policy **S**ampling), a paradigm that reuses promising prefixes of historical rollouts in RLVR training. PROS appends these self-generated partial rollouts to the original queries to form *Augmented Queries*, which are then used as regular training inputs in subsequent iterations, thereby reducing redundant computation. To select training batch from augmented queries, PROS adopts a hierarchical Bayesian model to estimate their pass rates and prioritize those with the highest reward uncertainty. Experiments across diverse settings show that PROS consistently improves training efficiency and achieves higher accuracy than strong baselines. These results highlight PROS as a practical path toward scalable and compute-efficient RLVR.

## 1 Introduction

Large reasoning models (LRMs) have achieved remarkable progress on complex tasks, especially in code generation and mathematical problem solving, in some cases even surpassing average human performance (OpenAI et al., 2024; Shao et al., 2024; DeepSeek-AI et al., 2025; Team et al., 2025b; Lambert et al., 2025). A key driver behind these advances is *Reinforcement Learning with Verifiable Rewards* (RLVR), where an LRM is optimized as a policy that generates chain-of-thoughts (CoTs) (Wei et al., 2023) as trajectories and receives binary rewards on final answers from deterministic verifiers. This paradigm enables models to improve reasoning abilities via supervision from verifiable outcomes, offering a scalable path for learning from self-exploration.

Despite its promise, the RLVR paradigm is constrained by heavy reliance on on-policy rollout generation. Each training iteration of a RLVR algorithm consists of four steps: *Select* queries from a given dataset as a training batch, *Generate* several rollouts based on selected queries with the policy, *Verify* these rollouts with a deterministic verifier to obtain binary rewards, and *Update* parameters of the policy to increase the likelihood of high-reward actions via policy-gradient methods such as PPO (Schulman et al., 2017) or GRPO (Shao et al., 2024). As models tackle increasingly complex problems, the length of their chain-of-thoughts (CoTs) grows accordingly. Consequently, the cost of online rollout generation escalates and soon dominates the overall training time, making it one of the principal obstacles to further scaling RLVR.

Our empirical analysis reveals that, for a given query, independently sampled reasoning trajectories can be naturally organized into a tree structure like a branching search over the answer space as shown in Figure 1. Although their final answers may diverge, the early steps often share similar lines of reasoning. This observation indicates substantial redundancy: by reusing the prefixes of historical rollouts in the previous iteration, we can avoid repeatedly generating these near-duplicate initial steps and thereby saving a significant amount of computation.

Based on this observation, we introduce PROS (**P**refix **R**euse for **O**n-policy **S**ampling), a paradigm designed to make RLVR more compute-efficient for further scaling. In each training iteration, PROS

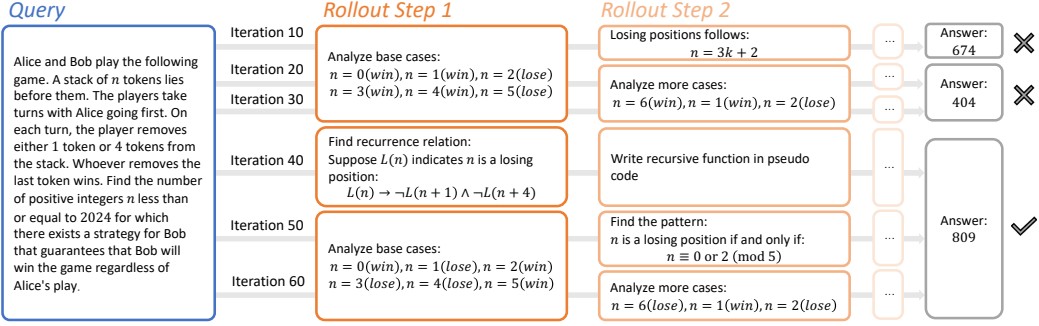

Figure 1: Rollouts for a query from different training iterations can naturally be organized into a tree structure, where their reasoning trajectories are highly similar in early steps and gradually diverge.

identifies high-quality prefixes of generated rollouts using readily available signals such as token-level entropy or value estimation from a critic. These rollout prefixes are then appended to their original queries to construct *Augmented Queries*. An augmented query preserves the task semantics of the original query while adding partial reasoning steps produced by the policy itself. It can be fed back into subsequent training iterations as a prompt, shortening on-policy generated reasoning steps required and thus reducing overall training compute. Beyond efficiency, PROS also enables a form of cross-iteration search pruning: it allows the policy to exploit those high-quality initial steps while avoiding useless early dead ends, thereby steering subsequent reasoning towards promising directions. This reallocates training compute to meaningful exploration in later reasoning steps, enabling the policy to acquire richer reasoning patterns.

As training progresses, more augmented queries are generated, and the dataset expands into a two-layer tree-structured hierarchy: each original query serves as a parent, and its derived augmented queries serve as children. A central challenge is selecting which augmented queries to train on. Bae et al. (2025) give a principle that queries with the highest reward variance exhibit the greatest learnability and provide the most informative training signals. Motivated by this, we instantiate a hierarchical Bayesian model over augmented queries, estimate per-query pass rate from historical reward statistics and prioritize those queries with high reward uncertainty (i.e. pass rate close to 0.5) (Hong et al., 2022a;b).

Essentially, PROS can be seamlessly integrated into most policy-gradient RLVR algorithms as a plugin. With the main algorithm logic unchanged, it only adds two additional steps, augmented query construction (Section 3) and augmented query selection (Section 4), both of which incur negligible computational overhead. In summary, this paper makes the following contributions:

- We propose **PROS**, a simple and general training paradigm that reuses high-value historical prefixes as Augmented Queries, reducing redundant generation and enhance policy exploitation.

- We further introduce a selection mechanism over all augmented queries that targets at the most valuable one for training, which is implemented via pass rate estimation by a hierarchical Bayesian model.

- PROS consistently outperforms strong baselines across diverse settings, achieving the best performance–cost trade-offs. Integrated as a plugin, it can further raise the upperbound of both PPO and GRPO, yielding an average improvements of +3.96 and +6.21 points on AIME24 and AMC23, respectively.

## 2 TRAINING BOTTLENECK ANALYSIS

**RLVR Basics** We begin by reviewing the process of *Reinforcement Learning with Verifiable Rewards* (RLVR) for training large reasoning models. For simplicity, we describe the setting with batch size 1, and denote the policy by $\pi_\theta$. In each training iteration, a query $q$ is selected from a training dataset, and $\pi_\theta$ autoregressively produces a chain of thought $y = [y_1, \ldots, y_T]$ conditioned on $q$. A deterministic verifier serves as an environment mapping $(q, y)$ to a binary reward

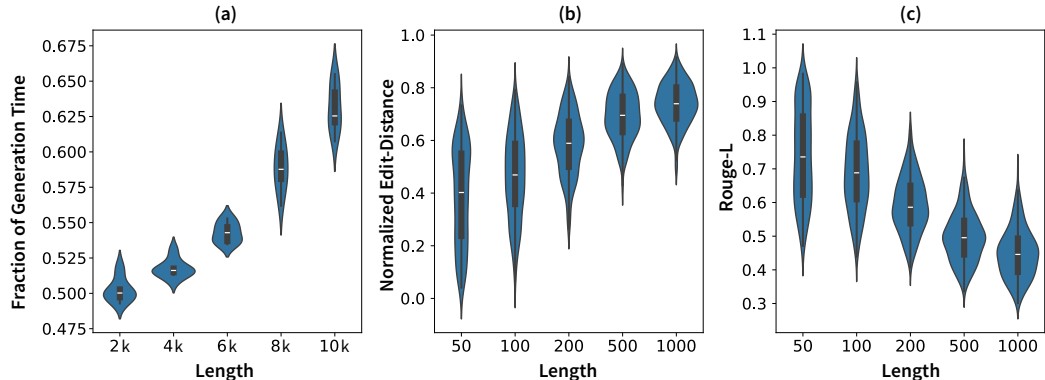

Figure 2: (a) Fraction of rollout generation time versus different max rollout length under fixed batch size and hardware. (b) Pairwise normalized edit distance versus prefix window size averaged over 64 rollouts per query. Higher value indicates lower similarity. (c) Self-rouge versus prefix window size averaged over 64 rollouts per query. Higher value indicates higher similarity.

$r = R(q, y) \in \{0, 1\}$, indicating whether the final answer matches the ground truth or satisfies a task-specific specification (e.g. passing all unit tests for code generation). The objective of RLVR finetuning is to optimize the policy parameters $\theta$ to maximize the expected rewards, i.e. $\max_\theta \mathbb{E}_{q; y \sim \pi_\theta(\cdot|q)}[R(q, y)]$, which is commonly implemented via policy-gradient methods like PPO or GRPO.

**Generation Share.** As both the scale of LRMs and the length of CoTs grow, on-policy rollout generation gradually dominates the training time of RLVR algorithms. We perform a preliminary measurement to quantify the generation share (i.e. the fraction of wall-clock time spent in rollout generation) under varying max rollout length $L_{max}$. As shown in Figure 2 (a), the generation share rises rapidly as the maximum generation length increases. Currently, the scale of LRMs have already approached hundreds of billions, with context lengths more than hundreds of thousands tokens. This trend highlights a scalability issue, underscoring the need for methods to alleviate generation costs.

**Redundancy in Early Reasoning Steps.** We empirically observe strong similarity at the beginning of different rollouts for the same query across training iterations. Their reasoning trajectories naturally unfold like a branching search: the early steps share similar setup, while later steps gradually diverge for different reasoning directions. A specific case is presented in Figure 1. To further quantify this effect, we sample 64 independent rollouts per query on the DAPO-Train dataset (Yu et al., 2025) and measure pairwise similarity on truncated prefixes of different lengths. Concretely, we compute the average of two pairwise similarity metrics: (i) normalized edit distance (i.e. EditDist/TotalLen), and (ii) Rouge-L (Lin, 2004). As shown in Figure 2, shorter prefixes yield markedly higher similarity, indicating substantial redundancy in repeatedly generating near-duplicate initial reasoning steps.

**Wasted Computation in Early Dead Ends.** A second source of redundancy also follows from the tree structure of reasoning trajectories. Correct solution trajectories are sparse in the reasoning space. Naive multi-sampling may therefore start on branches that are truly dead ends. Subsequently, their remaining suffix contribute little training value. For example, in Figure 1, rollouts in iterations 10 to 30 make a mistake at the first step and still consume long suffixes. This unnecessary dead-end cost compounds as model scale and sequence length grow.

## 3 AUGMENTED QUERY CONSTRUCTION

### 3.1 OVERVIEW

Building on the observations from Section 2, we introduce **PROS** (**P**refix **R**euse for **O**n-policy **S**ampling), a paradigm designed to mitigate redundant generation and wasted computation in RLVR.

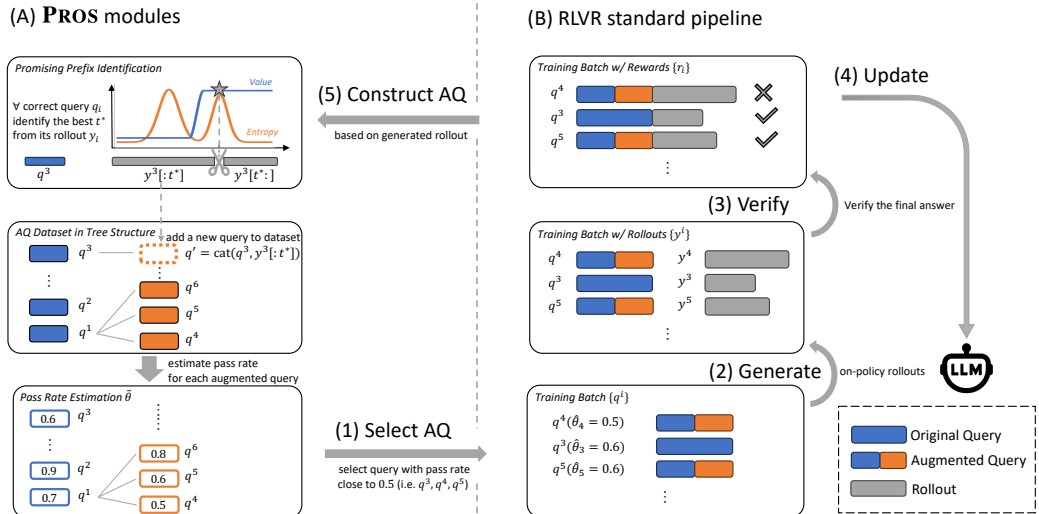

Figure 3: Overview of **PROS**. The right-hand side follows standard RLVR algorithms, while the left-hand side is the augmented query modules introduced by PROS. (1) Estimate the pass rate $\tilde{\theta}$ of each query and select those with highest uncertainty; (2) Generate rollouts $\{y^i\}$ for selected queries $\{q^i\}$; (3) Verify rollouts against ground truths; (4) Update the policy with rollouts and rewards; (5) Construct augmented queries by appending high-quality rollout prefixes to original queries (e.g. $q' = \mathrm{concat}(q^3, y^3[: t^*])$) for future reuse.

Concretely, in each training iteration, after the rollouts are generated and verified, we identify high-quality prefixes within them. We then append these valuable prefixes to their original queries to create new *Augmented Queries* (AQ), which are then added to the training set as new query instances for subsequent training iterations. Over time, this augmentation process expands the training data into a two-layer tree-structured dataset: each original query serves as a parent node, and its derived augmented queries become child nodes.

During training, augmented queries are treated equivalently as normal queries. The policy generates continual reasoning trajectories conditioned on both the historical prefix and the original query. The reinforcement learning update then proceeds over the collected continuations as in standard RLVR. The only difference is that both credit assignment and gradient update are conducted only on the newly generated continuations without the reused historical prefixes. This way, the policy is reinforced for how it continues from the prefix, preventing overfitting to the previously generated reasoning steps. Figure 3 illustrates the overall PROS pipeline.

Notably, an augmented query preserves the task semantics of its original query, only providing unverified partial reasoning steps generated by the policy itself. It does not reveal any final answer and retains the on-policy nature of the RLVR algorithm, in contrast to off-policy experience replay methods (Fedus et al., 2020; Schaul et al., 2016; Liang et al., 2021).

## 3.2 PROMISING PREFIX IDENTIFICATION

A crucial question in PROS is identifying which rollout prefixes are worth reusing. We employ a heuristic strategy based on both policy uncertainty and value signals, along with a length-based constraint, to construct augmented queries that are both informative and efficient.

We consider two signals to estimate a prefix's potential: uncertainty-based signal and value-based signal. As the uncertainty signal, we use the token-level entropy along the rollout, which serves as a proxy for uncertainty. A higher entropy indicates that the policy was less certain about the next steps, suggesting the prefix lies in an unexplored and informative region of the reasoning space. Such prefixes are candidates for reuse because they encourage exploration and mitigate over-confidence (Cui et al., 2025; Wang et al., 2025). Using entropy is almost free without extra computational overhead, since the RLVR algorithm already requires a forward pass to compute the log-probability at every timestep.

Besides entropy, we also leverage the readily available value function learned by critic model for those actor-critic methods (e.g. PPO). Prefixes with higher predicted value are more likely to lead to a correct final answer. So reusing them can help to exploit and focus computational costs on promising reasoning directions.

In practice, we reuse the prefix $y[:t^*] = [y_0, \ldots, y_{t^*}]$ with the highest token-level entropy from each correct rollout $y$, i.e. $t^* = \arg\max_{t \in [0,T)} \text{Entropy}(\pi(\cdot|y_{<t}, q))$. When the underlying RLVR algorithm is actor–critic, we additionally use the value function as a filter, restricting $t^*$ to the top 10% of timesteps with the highest predicted values. In addition to these signals, we also impose simple length constraints to seek a balance between exploitation and exploration. Specifically, we limit the range of $t^* \in [\frac{1}{4}T, \frac{3}{4}T)$. Although coarse, this constraint ensures that the reused prefix yields a non-trivial reduction in generation time, while leaving sufficient room for the policy exploration without giving away too much of the solution.

# 4 AUGMENTED QUERY SELECTION

## 4.1 OVERVIEW

As the augmented query dataset grows continuously across iterations, selecting a valuable training batch from a large amounts of AQs is non-trivial. Prior studies suggest that training is most effective when queries fall into an intermediate difficulty range rather than being trivially easy or impossibly hard (Yu et al., 2025; Razin et al., 2025; Vygotsky, 1978). Furthermore, Bae et al. (2025) theoretically demonstrates that queries with rollout pass rate close to $0.5$ exhibit the strongest learnability signal. Both Yu et al. (2025) and Bae et al. (2025) adopt an online filtering mechanism, removing queries whose pass rates are too high or too low *after* on-policy rollout generation. However, such online filtering requires costly rollout generation for every query. To reduce this overhead, we instead estimate pass rates of augmented queries from historical observations on rewards via Bayesian inference, and we prioritize those queries with estimated pass rates near $0.5$.

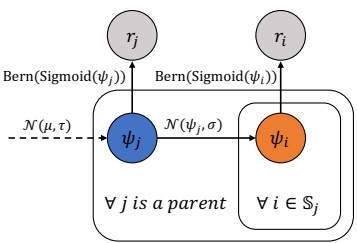

Figure 4: Overview of graphical model. $r$ indicates reward of each query, following a Bernoulli controlled by log odds $\psi$. $\psi_i$ of each augmented query $i$ is conditionally Gaussian centered on its parent's $\psi_j$. $\mathbb{S}_j$ indicates the children set of node $j$

As stated before, the augmented query dataset forms a tree-structured hierarchy, where each original query serves as a parent while its derived augmented queries serve as children. To leverage the relationship between original query and its derived augmented queries, we initialize a two-layer logit–normal Bayesian model for pass rate estimation (Hong et al., 2022a;b). Let pass rate $\theta = \text{sigmoid}(\psi) = (1 + \exp(-\psi))^{-1}$, we suppose the log-odds of each parent $par$ follows $\psi_{par} \sim \mathcal{N}(\mu, \tau^2)$ and the log-odds of its children $i$ is conditionally Gaussian, $\psi_i \mid \psi_{par} \sim \mathcal{N}(\psi_{par}, \sigma^2)$. Each time a query is used for generating a rollout, a binary reward $r \sim \text{Bern}(\theta)$ is then observed, indicating whether the rollout passes verification. The graphical model is presented in Figure 4.

This formulation encodes the inductive bias that an augmented query solves the same underlying problem as its parent but begins from an intermediate reasoning state, so a child's pass rate should be correlated with that of its parent. The hierarchical structure enables information sharing between related queries and improves statistical efficiency.

Building on the Bayesian model above, we can infer the posterior distribution of latent log-odds by incorporating historical reward observations. Posterior samples $\{\tilde{\psi}\}$ are then drawn for each query, and the corresponding pass rates $\tilde{\theta} = \sigma(\tilde{\psi})$ are estimated for training batch selection.

## 4.2 BAYESIAN INFERENCE WITH PÓLYA–GAMMA AUGMENTATION

In order to sample from the joint posterior, we derive the full conditional distributions for both parents and children, which allows Gibbs sampling. Specifically at iteration $t$, let $H_i = (n_i, s_i)$ denote the reward history of query $i$, where $n_i$ is the number of times the query being selected and $s_i$ the number of success times of its corresponding rollouts. Suppose the parent set of nodes is $\mathbb{F}$, and the children set of node $j$ is $\mathbb{S}_j$. By Bayes' rule, the joint posterior admits a hierarchical chain

factorization:

$$p(\{\psi\}|H) \propto \prod_{j\in\mathbb{F}} \left[ \underbrace{p(\psi_j)}_{\text{prior of parent}} \times \underbrace{p(H_j|\psi_j)}_{\text{likelihood of parent}} \times \prod_{i\in\mathbb{S}_j} \underbrace{p(\psi_i|\psi_j)}_{\text{prior of child}} \times \underbrace{p(H_i|\psi_i)}_{\text{likelihood of child}} \right]$$

A key challenge in posterior calculation above is the non-conjugacy of the Bernoulli likelihood and Gaussian prior. We address this by introducing Pólya–Gamma (PG) auxiliary variables following Polson et al. (2013); Dumitrascu et al. (2018), which render the calculation tractable.

We first derive the conditional posterior $p(\psi_i|H_i, \psi_{par})$ of a child $i$, given its parent $\psi_{par}$ and history $H_i = (n_i, s_i)$,

$$p(\psi_i|H_i,\psi_{par}) \propto p(\psi_i|\psi_{par}) \cdot p(H_i|\psi_i) = p(\psi_i|\psi_{par})2^{-n_i} \int_0^\infty \exp(\kappa\psi_i - \frac{\omega_i\psi_i^2}{2})p_{PG}(\omega_i|n_i,0)d\omega_i$$

where $\kappa = s_i - \frac{n_i}{2}$ and $\omega_i \sim PG(n_i, 0)$. The identity follows directly from Polson et al. (2013, Theorem 1). By introducing an auxiliary variable $\omega_i$, the conditional posterior $p(\psi_i|H_i, \psi_{par}, \omega_i)$ becomes Gaussian and thus tractable, given that the prior $p(\psi_i|\psi_{par})$ is Gaussian. In particular, we have the following proposition:

**Proposition 4.1.** *Given the log-odds of parent $\psi_{par}$ and the history $H_i = (n_i, s_i)$,*

$$\omega_i|\psi_i,H_i \sim \mathrm{PG}(n_i,\psi_i) \quad \psi_i|\omega_i,H_i,\psi_{par} \sim \mathcal{N}(m,V) \text{ where } V = \frac{1}{\sigma^{-2}+\omega_i}, m = V\cdot(\sigma^{-2}\psi_{par}+\kappa)$$

Similarly, we introduce a PG variable $\omega_{par}$ for each parent $\psi_{par}$ and derive its conditional posterior with the same augmentation method:

**Proposition 4.2.** *Given the log-odds of all children $\{\psi_k\}_k^{\mathbb{S}_{par}}$ and the history $H_{par} = (n_{par}, s_{par})$,*

$$\omega_{par}|\psi_{par},H_{par} \sim \mathrm{PG}(n_{par},\psi_{par}) \quad \psi_{par}|\omega_{par},H_{par},\{\psi_k\}_k^{\mathbb{S}_{par}} \sim \mathcal{N}(m,V)$$

*where $V = (\tau^{-2} + \omega_{par} + |\mathbb{S}_{par}|\sigma^{-2})^{-1}$, $m = V\cdot(\tau^{-2}\mu + \sigma^{-2}\sum_k\psi_k + \kappa_{par})$*

Proposition 4.1 and 4.2 together define a simple Gibbs sampler for the joint posterior $p(\{\psi\}, \{\omega\}|H)$, from which we can efficiently sample $\tilde{\psi}$ and estimate pass rates $\tilde{\theta}$ for all augmented queries. The overall estimation process is presented in Algorithm 1. Detailed proofs in this section are presented in Appendix B.

---

**Algorithm 1** Pass Rate Estimation for Augmented Query

---

**Input:** Log-odds of all queries from previous iteration $\{\psi\}$; History of all queries $\{(n,s)\}$; Number of Gibbs sweeps $G$; Model Hyperparameters $\mu,\tau,\sigma$

**for** $t=1$ **to** $G$ **do**
  // One Gibbs sweep
  **foreach** *parent* $par\in\mathbb{F}$ **do**
    **foreach** *child* $i\in\mathbb{S}_{par}$ **do**
      // Posterior sampling based on Proposition 4.1
      sample $\omega_i \sim PG(n_i,\psi_i)$;
      $V \leftarrow (\sigma^{-2}+\omega_i)^{-1}, m \leftarrow V\cdot(\sigma^{-2}\psi_{par}+\kappa)$, sample $\psi_i \sim \mathcal{N}(m,V)$
    **end**
    // Posterior sampling based on Proposition 4.2
    sample $\omega_{par} \sim PG(n_{par},\psi_{par})$; $V \leftarrow (\tau^{-2}+\omega_{par}+|\mathbb{S}_{par}|\sigma^{-2})^{-1}, m \leftarrow V\cdot(\tau^{-2}\mu+\sigma^{-2}\sum_k\psi_k + \kappa_{par})$, sample $\psi_{par} \sim \mathcal{N}(m,V)$
  **end**
**end**

---

### 4.3 EXPONENTIAL FORGETTING FOR NON-STATIONARY ENVIRONMENT

As training progresses, the policy evolves continuously, which leads to gradual shifts in the pass rate of queries. Therefore, we introduce an exponential forgetting mechanism. At each iteration,

Table 1: Comparison between PROS and other baselines under different settings. `#Tokens` denotes the average number of tokens generated per iteration (in millions), and `#Time` denotes the average GPU wall-clock time per iteration (in minutes).

| | DAPO-Train | | | | AIME-Old | | | |
|---|---|---|---|---|---|---|---|---|
| | AIME24 | AMC23 | #Tokens | #Time | AIME24 | AMC23 | #Tokens | #Time |
| PPO | 28.23 | 73.63 | 10.24 | 9.17 | 31.35 | 57.85 | 4.39 | 6.06 |
| w/ Dynamic Sampling | 28.85 | 66.69 | 11.07 | 17.62 | 33.96 | **69.21** | 9.10 | 40.04 |
| w/ Experience Replay | 30.10 | 73.02 | 8.93 | 8.98 | 31.35 | 57.85 | 3.61 | 5.88 |
| w/ Priority Sampling | 31.87 | 73.25 | 11.35 | 9.76 | 31.46 | 60.82 | 4.15 | 5.91 |
| w/ PROS-ablation | 31.35 | 73.93 | 7.92 | 9.29 | 31.87 | 65.09 | 6.93 | 8.89 |
| w/ PROS | **33.23** | **78.20** | 8.49 | 9.77 | **34.27** | 65.62 | 5.00 | 8.09 |
| GRPO | 29.58 | 73.40 | 9.27 | 6.58 | 31.04 | 59.98 | 3.58 | 3.77 |
| w/ Dynamic Sampling | 31.15 | 76.68 | 11.16 | 19.45 | 29.58 | **68.90** | 9.22 | 45.38 |
| w/ Experience Replay | 29.90 | 74.62 | 7.60 | 6.36 | 31.15 | 59.98 | 3.14 | 4.14 |
| w/ Priority Sampling | 30.73 | 73.70 | 9.55 | 6.87 | 33.44 | 60.14 | 3.88 | 4.33 |
| w/ PROS-ablation | 30.52 | 75.91 | 7.87 | 7.00 | 33.44 | 63.41 | 5.88 | 6.26 |
| w/ PROS | **34.17** | **78.28** | 8.46 | 7.57 | **34.38** | 67.53 | 5.84 | 6.69 |

Table 2: Comparison between PROS and other baselines on Qwen3-4B trained with AIME-Old.

| | AIME24 | AMC23 | #Time |
|---|---|---|---|
| PPO | 21.25 | 52.59 | 5.83 |
| w/ dynamic | 25.73 | 61.19 | 10.83 |
| w/ replay | 23.75 | 53.73 | 5.81 |
| w/ prior | 24.90 | 60.44 | 5.86 |
| w/ PROS | **27.40** | **62.12** | 6.21 |

Table 3: Ablation study on hyperparameters by varying $\sigma$ and $\lambda$. PROS consistently surpasses Vanilla (59.98 on AMC23; 31.04 on AIME24).

| $\lambda = 0.99$ (default) | | | $\sigma = 0.3$ (default) | | |
|---|---|---|---|---|---|
| $\sigma$ | AMC23 | AIME24 | $\lambda$ | AMC23 | AIME24 |
| 0.10 | 64.25 | 34.79 | 0.95 | 65.62 | **35.94** |
| 0.20 | 62.42 | **36.98** | 0.99 | **67.53** | 34.38 |
| 0.30 | **67.53** | 34.38 | 0.995 | 66.77 | 33.44 |

we scale down the historical statistics $s_i$ and $n_i$ of every augmented query by a forgetting factor $\lambda \in (0, 1)$ before incorporating new observations. This exponential forgetting ensures that more recent rewards exert greater influence on posterior updates, allowing the sampler to adapt to policy improvements.

## 5 EXPERIMENTS

We conduct main experiments on the Qwen3-8B model, trained using PPO (Schulman et al., 2017) and GRPO (Shao et al., 2024). For evaluation, we report *Pass@1* on two benchmarks: AIME 2024 and AMC 2023. To mitigate variance, we report averages over 32 and 16 independent runs on these datasets, respectively, following Hochlehnert et al. (2025).

**Training configuration.** We adopt two math reasoning datasets as training corpora: *DAPO-Train* is a large and diverse corpus covering broad domains of math problems(Yu et al., 2025); *AIME-Old* consists of all AIME problems prior to 2024, which is more relevant to the AIME24 benchmark. We train for 400 and 300 iterations on these datasets, respectively. At each iteration, we generate 8 rollouts per query. The batch size is 512 with mini-batch size 64, yielding 8 gradient updates per PPO epoch. The maximum response length is set to 6144 tokens.

**Baselines.** We compare our proposed **PROS** against several strong baselines: (1) *Vanilla* is the standard PPO/GRPO training algorithm. (2) *Dynamic Sampling* (Yu et al., 2025) constructs training batches by filtering queries whose on-policy rollouts are either all correct or all incorrect. (3) *Priority Sampling* (Team et al., 2025b;a) tracks the historical pass rate $\tilde{\theta}$ of each query and samples proportionally to $1 - \tilde{\theta}$. (4) *Experience Replay* augment PPO/GRPO with a replay buffer with replay ratio set to $1/8$. We also apply truncated importance sampling following ACER (Wang et al., 2017). (5) PROS-*ablation*: a variant of PROS with query being randomly sampled, isolating the effect of proposed augmented query selection mechanism. Additional details are provided in Appendix C.

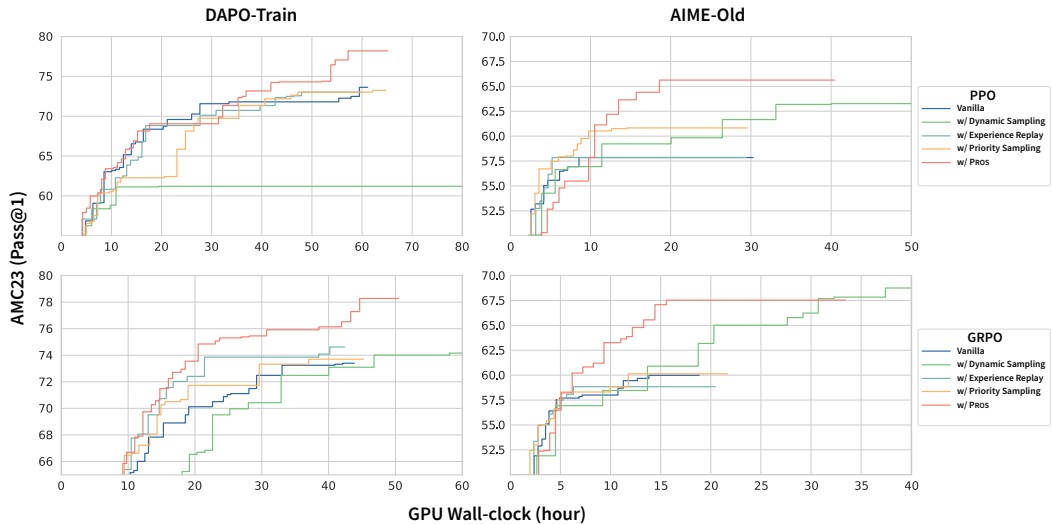

Figure 5: Training efficiency comparison among different methods. The $x$-axis denotes the training time (GPU hours), and the $y$-axis reports the best performance achieved up to that time.

## 6 MAIN RESULTS

**Overall performance.** Table 1 summarizes the comparison between PROS and other baselines. PROS delivers consistent improvements across all four settings. Under PPO trained on DAPO-Train, it outperforms the vanilla baseline by $+5$ and achieves the best performance on both AIME24 and AMC23. Comparable gains are observed under GRPO. These results validate our hypothesis: reusing promising prefixes enhances exploitation, thereby improving overall performance. In contrast, *Experience Replay* improves efficiency but yields performance comparable to vanilla. *Dynamic Sampling* achieves strong results, particularly on AMC23, but requires $2$–$4\times$ more wall-clock time than PROS due to repeated rejection sampling. By comparison, PROS attains strong performance with only modest computational overhead. Most of the extra cost comes from its length scaling behavior (see below), which yields longer CoTs. To isolate the influence of augmented query sampling, we compare PROS with the PROS-*ablation* variant. The results show that prefix reuse alone already improves compute efficiency and achieves better performance. Moreover, incorporating the proposed augmented query selection mechanism consistently provides additional gains. We also provide an additional experiments on Qwen3-4B in Table 2, which exhibit similar trends.

**Performance-Cost trade-offs.** To further assess the compute efficiency of different methods, we analyze the performance growth with respect to GPU wall-clock time in Figure 5. It is shown that PROS consistently exhibits better efficiency under different settings. Compared to dynamic sampling, which achieves competitive performance but at prohibitive cost, PROS provides a significantly more favorable balance between efficiency and performance.

**Influence of Hyper-parameters.** We conduct an ablation study to evaluate the robustness of PROS with respect to key hyperparameters. In the Bayesian model (Section 4.1), the child prior $\psi_i \mid \psi_{\mathrm{par}} \sim \mathcal{N}(\psi_{\mathrm{par}}, \sigma^2)$ introduces a variance parameter $\sigma^2$ that controls similarity between parent and child nodes, while the temporal decay factor $\lambda$ governs the rate of forgetting in pass rate estimation. We train PROS with GRPO on the AIME-Old dataset, varying both $\sigma$ and $\lambda$. Table 3 reports Pass@1 results, and Figure 7 shows the estimation error of pass rates. The results show that PROS consistently brings improvements to vanilla GRPO across all hyperparameter settings, and the pass rate estimation remains robust, becoming increasingly accurate as training proceeds.

**Length scaling.** A key property of RLVR algorithms is their ability to benefit from increased reasoning lengths, which enables improved test-time scaling (Snell et al., 2024). We present the length scaling trends of different methods in Figure 6. Across all settings, the rollout length of PROS continues to scale up as the training goes. By contrast, vanilla PPO/GRPO, as well as their experi-

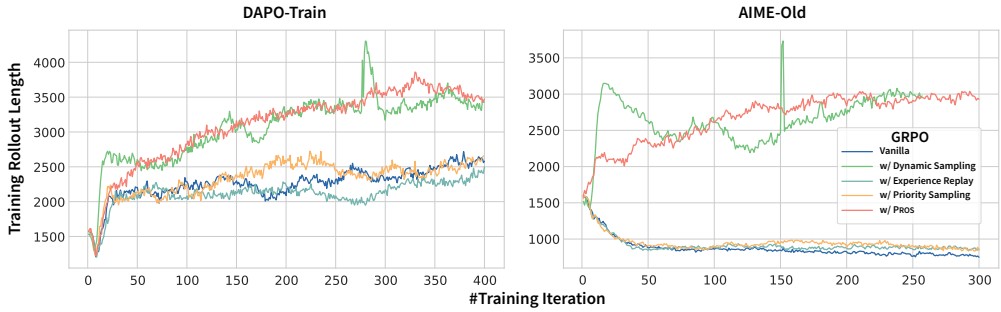

Figure 6: Length scaling trends of different methods under different settings.

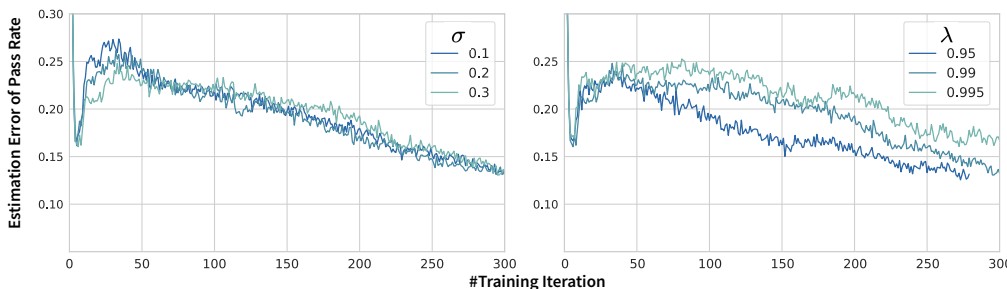

Figure 7: Average pass rate estimation error in PROS with different hyperparameters.

ence replay and prioritized sampling variants, suffer from length collapse on the AIME-Old dataset, limiting their ability to further exploitation.

## 7 RELATED WORK

Our proposed method focuses on improving efficiency of rollout generation in RLVR. In traditional reinforcement learning, a common approach to reduce the cost of on-policy rollout generation is experience replay (Fedus et al., 2020; Schaul et al., 2016; Liang et al., 2021). For instance, PPO typically reuses the same batch of rollouts for multiple training epochs. However, He et al. (2025) show that, due to the complexity of large language models, repeatedly reusing the same data quickly leads to overfitting and entropy collapse, which prevents RLVR from scaling effectively in the long run. Another line of related work reduces task difficulty by providing partial reference solutions as hints during training, which requires costly human annotation(Xi et al., 2024; Liu et al., 2025a;b). In contrast, our approach derives partial solutions directly from the model's own past rollouts, removing the need for external reference solutions in complex reasoning settings. Thirdly, our augmented query selection also draws inspiration from curriculum learning (Soviany et al., 2022; Narvekar et al., 2020; Wang et al., 2021), by dynamically selecting training queries whose difficulty best matches the current policy. Several concurrent studies also investigate online data selection in RLVR to improve performance (Sun et al., 2025; Bae et al., 2025; Zheng et al., 2025; Qu et al., 2025). Our method differs in that it is specifically designed for the hierarchical augmented dataset.

## 8 CONCLUSION

We presented **PROS**, a prefix-reuse paradigm for RLVR training that constructs augmented queries from historical rollout and leverages a hierarchical Bayesian model for uncertainty-aware selection. Experiments demonstrate its consistent improvements in both efficiency and accuracy compared to strong baselines, highlighting the promise of PROS for compute-efficient training of LRMs.

While effective, our current design of augmented queries is still simple, relying on entropy and value signals that may introduce bias. Future work could investigate richer or more principled criteria for prefix identification, explore adaptive integration with other forms of uncertainty estimation, and extend prefix reuse to broader domains beyond mathematical reasoning.

## 9 ACKNOWLEDGEMENT

This work was supported by Beijing Natural Science Foundation (L253001), Key Laboratory of Science, Technology and Standard in Press Industry (Key Laboratory of Intelligent Press Media Technology) and National Engineering Research Center of New Electronic Publishing Technologies. We appreciate the anonymous reviewers for their helpful comments. Xiaojun Wan is the corresponding author.

### ETHICS STATEMENT

This work focuses on improving the compute efficiency of reinforcement learning with verifiable rewards (RLVR) for large reasoning models. All experiments are conducted on publicly available datasets (AIME, AMC, DAPO) that do not contain sensitive personal information. Our approach does not introduce new data collection, and we adhere to the original licenses and intended use of these datasets.

We believe this work raises no additional ethical concerns beyond those already inherent in the study of large language models and reinforcement learning. We emphasize that our contributions are intended solely for research, and caution should be exercised when transferring these methods to real-world applications.

### REPRODUCIBILITY STATEMENT

All experiments in this paper are conducted on publicly available datasets (AIME, AMC, and DAPO), which can be readily accessed and downloaded. Complete proofs of the theoretical propositions are provided in Appendix B. Detailed descriptions of experimental settings, including training configurations, hyperparameters, and baseline implementations, are provided in Appendix C to facilitate replication. The full source code repository together with reproducibility scripts is included in the supplementary materials.

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

## A  THE USE OF LARGE LANGUAGE MODELS

Large language models (LLMs) were only used for editing and polishing the text of this paper, in order to improve clarity and fluency of presentation. They were not used for generating ideas, conducting experiments, analyzing results, or writing technical content.

## B  PROOFS OF SECTION 4.2

### B.1  PROBLEM SETUP

We first review the problem and our proposed Bayesian model. Selection on the AQ tree can be casted as a hierarchical multi-armed bandit: each parent arm corresponds to an original query and its children correspond to its derived augmented queries. Each arm (i.e. query) inherently has a success rate $\theta$. Pulling an arm yields a binary reward $r \sim \text{Bern}(\theta)$, indicating whether one generated rollout from the query is correct.

We propose a two-layer logit–normal Bayesian model for success rate estimation. Let success rate $\theta = \text{sigmoid}(\psi) = (1 + \exp(-\psi))^{-1}$. We suppose the log-odds of each parent $par$ follows $\psi_{par} \sim \mathcal{N}(\mu, \tau^2)$ and the log-odds of its children $i$ is conditionally Gaussian, $\psi_i \mid \psi_{par} \sim \mathcal{N}(\psi_{par}, \sigma^2)$.

The objective is to leverage the historical observations on rewards to derive the joint posterior conditioned on history, from which we can sample log-odds parameter $\tilde{\psi}$ of both parents and children and then estimate their success rates $\tilde{\theta} = \text{sigmoid}(\tilde{\psi})$.

Specifically at iteration $t$, let $H_i = (n_i, s_i)$ denote the reward history of query $i$, where $n_i$ is the number of the query being selected and $s_i$ the number of successes of its corresponding rollouts. Suppose the father set of nodes is $\mathbb{F}$, and the children set of node $j$ is $\mathbb{S}_j$. By Bayes' rule, the joint posterior admits a hierarchical chain factorization:

$$p(\{\psi\}|H) \propto \prod_{j \in \mathbb{F}} \left[ \underbrace{p(\psi_j)}_{\text{prior of parent}} \times \underbrace{p(H_j|\psi_j)}_{\text{likelihood of parent}} \times \prod_{i \in \mathbb{S}_j} \underbrace{p(\psi_i|\psi_j)}_{\text{prior of child}} \times \underbrace{p(H_i|\psi_i)}_{\text{likelihood of child}} \right] \tag{1}$$

In the following, we will apply Pólya–Gamma augmentation to render the posterior sampling tractable by sample from an augmented posterior $p(\{\psi\}, \{\omega\}|H)$, which has a single Gibbs sampler.

### B.2 Proofs of Proposition 4.1

Polson et al. (2013) gives the following lemma about PG distribution:

**Lemma B.1.** *(Pólya-Gamma identity) Let $p(\omega)$ denote the density of the random variable $\omega \sim PG(b, 0)$, $b > 0$. Then the following integral identity holds for all $a \in \mathbb{R}$:*

$$\frac{\exp(\psi)^a}{(1 + \exp(\psi))^b} = 2^{-b} \exp((a - b/2)\psi) \int_0^\infty \exp(-\omega\psi^2/2)p(\omega)d\omega$$

We first derive the conditional posterior $p(\psi_i|H_i, \psi_{par})$ of a child $i$, given its parent $\psi_{par}$ and history $H_i = (n_i, s_i)$,

$$\begin{aligned}
&p(\psi_i|H_i, \psi_{par}) \\
\propto& p(\psi_i|\psi_{par}) \cdot p(H_i|\psi_i) \\
=& p(\psi_i|\psi_{par}) \cdot p(s_i, n_i|\psi_i) \\
=& p(\psi_i|\psi_{par}) \cdot \frac{\exp(\psi_i)^{s_i}}{(1 + \exp(\psi_i))^{n_i}} \\
=& p(\psi_i|\psi_{par}) \cdot 2^{-n_i} \int_0^\infty \exp(\kappa\psi_i - \frac{\omega_i\psi_i^2}{2})p_{PG}(\omega_i|n_i, 0)d\omega_i
\end{aligned}$$

where $\kappa = s_i - \frac{n_i}{2}$ and $\omega_i \sim PG(n_i, 0)$. The identity follows directly from Lemma B.1. Now that we have the following identities:

$$p(\psi_i, \omega_i|H_i, \psi_{par}) \propto p(\psi_i|\psi_{par}) \cdot \exp(\kappa\psi_i - \frac{\omega_i\psi_i^2}{2})p_{PG}(\omega_i|n_i, 0)$$

$$p(\omega_i|\psi_i, H_i, \psi_{par}) = \frac{\exp(-\frac{\omega_i\psi_i^2}{2})p_{PG}(\omega_i|n_i, 0)}{\int_0^\infty \exp(-\frac{\omega_i\psi_i^2}{2})p_{PG}(\omega_i|n_i, 0)d\omega_i} = p_{PG}(\omega_i|n_i, \psi_i)$$

The second identity follows from the definition of Pólya-Gamma probability density. The full conditional posterior of $\psi_i$ then follows:

$$p(\psi_i|H_i, \psi_{par}, \omega_i) = p(\psi_i, \omega_i|H_i, \psi_{par})/p(\omega_i|H_i) \propto p(\psi_i|\psi_{par}) \cdot \exp(\kappa\psi_i - \frac{\omega_i\psi_i^2}{2})$$

Given that the prior $\psi_i|\psi_{par} \sim \mathcal{N}(\psi_{par}, \sigma^2)$, $\psi_i$'s full conditional posterior $p(\psi_i|H_i, \psi_{par}, \omega_i)$ is also a Gaussian:

$$\psi_i|H_i, \psi_{par}, \omega_i \sim \mathcal{N}(m, V), \text{ where } V = \frac{1}{\sigma^{-2} + \omega_i}, \quad m = V \cdot (\sigma^{-2}\psi_{par} + \kappa)$$

And the proof of Proposition 4.1 ends.

### B.3 PROOFS OF PROPOSITION 4.2

Similarly, we can also derive parents' conditional posteriors $p(\psi_{par}|H_{par}, \{\psi_k\}_k^{\mathbb{S}_{par}})$ by introducing PG variables:

$$p(\psi_{par}|H_{par}, \{\psi_k\}_k^{\mathbb{S}_{par}}) \propto p(\psi_{par}) \cdot p(s_{par}, n_{par}|\psi_{par}) \cdot \prod_{k \in \mathbb{S}_{par}} p(\psi_k|\psi_{par})$$

$$= \left( p(\psi_{par}) \cdot \prod_{k \in \mathbb{S}_{par}} p(\psi_k|\psi_{par}) \right) \cdot 2^{-n_{par}} \int_0^\infty \exp(\kappa_{par}\psi_{par} - \frac{\omega_{par}\psi_{par}^2}{2}) p_{PG}(\omega_{par}|n_{par}, 0) d\omega_{par}$$

where $\kappa_{par} = s_{par} - \frac{n_{par}}{2}$, $\omega_{par} \sim PG(n_{par}, 0)$. The identity also follows from Lemma B.1. Based on this, we can easily derive the following conditional posteriors:

$$p(\psi_{par}, \omega_{par}|H_{par}, \{\psi_k\}_k^{\mathbb{S}_{par}}) \propto \left( p(\psi_{par}) \cdot \prod_{k \in \mathbb{S}_{par}} p(\psi_k|\psi_{par}) \right) \exp(\kappa_{par}\psi_{par} - \frac{\omega_{par}\psi_{par}^2}{2})$$
$$\cdot p_{PG}(\omega_{par}|n_{par}, 0)$$

$$p(\omega_{par}|\psi_{par}, H_{par}) = \frac{\exp(-\frac{\omega_{par}\psi_{par}^2}{2}) p_{PG}(\omega_{par}|n_{par}, 0)}{\int_0^\infty \exp(-\frac{\omega_{par}\psi_{par}^2}{2}) p_{PG}(\omega_{par}|n_{par}, 0)} = p_{PG}(\omega_{par}|n_{par}, \psi_{par})$$

$$p(\psi_{par}|\omega_{par}, H_{par}, \{\psi_k\}_k^{\mathbb{S}_{par}}) \propto \left( p(\psi_{par}) \cdot \prod_{k \in \mathbb{S}_{par}} p(\psi_k|\psi_{par}) \right) \exp(\kappa_{par}\psi_{par} - \frac{\omega_{par}\psi_{par}^2}{2})$$

Given that prior $\psi_{par} \sim \mathcal{N}(\mu, \tau^2)$, $\psi_k|\psi_{par} \sim \mathcal{N}(\psi_{par}, \sigma^2)$, $\psi_{par}$'s full conditional posterior is also a Gaussian:
$$\psi_{par}|\omega_{par}, H_{par}, \{\psi_k\}_k^{\mathbb{S}_{par}} \sim \mathcal{N}(m, V)$$
where $V = \frac{1}{\tau^{-2} + \omega_{par} + |\mathbb{S}_{par}|\sigma^{-2}}$, $m = V \cdot (\tau^{-2}\mu_0 + \sigma^{-2}\sum_k \psi_k + \kappa_{par})$. And the proof of Proposition 4.2 ends.

## C EXPERIMENTAL SETTINGS

The main experiments are conduct on PPO and GRPO with implementation in verl[1]. At each training iteration, we generate 8 rollouts per query with temperature set to 1.0. We use a batch size of 512 and a mini-batch size of 64, yielding 8 gradient updates per PPO epoch. The maximum response length is set to 6144 tokens. We adopt the clip-higher strategy ($\epsilon_{high} = 0.28$) and the overlong reward shaping ($L_{cache} = 1024$) introduced by Yu et al. (2025), but do not apply additional KL regularization or entropy loss. Specifically for PPO, we adopt the decoupled-GAE and length-adaptive GAE proposed in VAPO (Yue et al., 2025). We also conduct value pretraining for twenty iterations following VAPO. For dynamic sampling baseline, we reuse the implementation code in verl. For prioritized sampling, we tracks the number of rollouts being generated for each query and the number of success times within these rollouts to calculate the pass rates. Similar to our proposed PROS, we also adopt a exponential decay of $\lambda = 0.9$. For experience replay, we only reuse the rollouts from the last iteration to avoid large gap. In specific, we use $64 \times 8$ rollouts from the replay buffer, while the rest of $(512 - 64) \times 8$ rollouts are generated by the current policy. The truncation threshold for truncated importance sampling is set to 10 following Wang et al. (2017). For PROS, we adopt $\mu = 0, \tau = 1.5$ to ensure a near uniform prior of pass rate $\theta$. The exponential discounting factor $\lambda = 0.99$ and the variance of children prior $\sigma = 0.3$ by default. We also apply a diversity regularization in the selection stage that each query appears at most one time within any consequential K training iterations. To ensure a fair comparison with respect to training efficiency and GPU wall-clock, we apply identical engineering hyperparameters to all methods, such as `gpu_memory_utilization` for inference engine, `max_token_len_per_gpu` for dynamic batching, etc.

---

[1] https://github.com/volcengine/verl

Table 4: Comparison of Pass@32 between PROS and other baselines of Qwen3-8B trained on DAPO-Train.

|  | AIME24 | AMC23 | #Tokens | #Time |
|---|---|---|---|---|
| PPO | 57.67 | 90.03 | 10.24 | 9.17 |
| w/ Dynamic Sampling | 63.89 | 90.69 | 11.07 | 17.62 |
| w/ Experience Replay | 63.15 | 90.05 | 8.93 | 8.98 |
| w/ Priority Sampling | 64.55 | 91.41 | 11.35 | 9.76 |
| w/ PROS-ablation | 62.64 | **92.13** | 7.92 | 9.29 |
| w/ PROS | **66.05** | 90.87 | 8.49 | 9.77 |
| GRPO | 59.14 | 90.62 | 9.27 | 6.58 |
| w/ Dynamic Sampling | 63.82 | 91.09 | 11.16 | 19.45 |
| w/ Experience Replay | 63.48 | 91.05 | 7.60 | 6.88 |
| w/ Priority Sampling | 64.77 | 91.02 | 9.55 | 6.87 |
| w/ PROS-ablation | **65.44** | **91.24** | 7.87 | 7.00 |
| w/ PROS | 65.01 | 90.54 | 8.46 | 7.57 |

Table 5: More ablations of PROS with Qwen3-4B trained on AIME-Old.

|  | AIME24 | AMC23 | #Time |
|---|---|---|---|
| PPO | 21.25 | 52.59 | 5.83 |
| w/ dynamic | 25.73 | 61.19 | 10.83 |
| w/ replay | 23.75 | 53.73 | 5.81 |
| w/ prior | 24.90 | 60.44 | 5.86 |
| w/ PROS (value-based prefix identification) | 24.27 | **65.32** | 5.97 |
| w/ PROS (entropy-based prefix identification) | 24.68 | 63.79 | 6.33 |
| w/ PROS (reward variance selection) | 25.72 | 61.81 | 6.07 |
| w/ PROS | **27.40** | 62.12 | 6.21 |

## D  ADDITIONAL EXPERIMENTS

Here we present the additional experiments required by reviewers.

Table 4 presents the Pass@32 comparison between PROS and other baselines.

Table 5 presents additional ablation studies on PROS. The entropy-based prefix identification variant selects the prefix with the highest token-level entropy (which is the same selection heuristic used in GRPO w/ PROS) for future reuse. The value-based prefix identification variant selects the prefix with the highest value according to the critic. Both of these variants outperform the baseline by a large margin, demonstrating the effectiveness and generalization of our proposed prefix reuse mechanism. The reward variance selection variant replaces the proposed Bayesian-based selection mechanism with an $\epsilon$-greedy method, which greedily selects augmented queries with the highest reward variance from history, while still maintaining an $\epsilon$ mechanism to ensure exploration.

