# OpenReview forum: "PROS: Towards Compute-Efficient RLVR via Rollout Prefix Reuse"
_ICLR.cc/2026/Conference — ICLR 2026 Poster_

### Official Review · Reviewer_NFuG · 2025-10-19

**Soundness:** 2
**Presentation:** 3
**Contribution:** 2
**Rating:** 6
**Confidence:** 4

**Summary:**

This paper introduces PROS, an RLHF training paradigm that reuses correct-prone trajectories to augment raw prompts, thereby improving exploration toward correct responses. PROS employs a Bayesian model to prioritize training on augmented prompts with high uncertainty, via token-level or progress-based values. Experimental results show that PROS achieves faster convergence and higher accuracy compared to baseline methods.

**Strengths:**

+ It tackles an important problem in RLHF.
+ The design is conceptually clear and practical to implement.
+ The paper is overall well-written.
+ Evaluations demonstrate good speedups and accuracy gains.

**Weaknesses:**

- Evaluations are limited to only two datasets and miss some ablation studies.
- The methodological novelty is moderate; the core ideas resemble existing works such as TreeRPO[1].

**Questions:**

Thank you for submitting your work. The paper is clearly presented and provides a reasonable solution to improve RLHF efficiency. However, the contribution would be stronger with more comprehensive experiments and deeper analysis. Below are specific comments and questions:

- Q1: How does PROS perform on other datasets, such as the code generation task or MATH500? Using only AIME24 and AMC23 limits the generality of the results.

- Q2: Reusing correct-prone trajectories may improve sample efficiency but risks reducing response diversity. Can the authors discuss potential strategies to mitigate this effect?

- Q3: The ablation study on Bayesian Inference (BI) for prompt selection is appreciated. Could the paper also compare simpler alternatives (e.g., variance-based scoring) to justify whether BI offers sufficient benefit to warrant its added complexity?

- Q4: In Figure 5(a), some baselines (e.g., Vanilla PPO) appear to have been early-stopped before convergence. Could the paper clarify the stopping criteria?

- Q5: Tables 2 and 3 include useful hyperparameter studies. It would be better also to show how the baseline methods perform under the same hyperparameter variations for a fairer comparison.

Reference

[1] TreeRPO: Tree Relative Policy Optimization, arXiv:2506.05183.

---

> ### Author Response · Authors · 2025-11-15
>
> > the core ideas resemble existing works such as TreeRPO
>
> Thank you for your comment. While TreeRPO proposes a variant of GRPO that leverages the tree structure of rollouts for better credit assignment, our work presents a novel training framework that can be applied to most RLVR algorithms (including PPO, GRPO, REINFORCE++, etc.). We are curious about the specific arguments regarding the similarity between TreeRPO and our approach. Additionally, we would like to kindly point out that TreeRPO is also submitted to ICLR 2026, and should be considered a concurrent work.
>
> We are happy to cite TreeRPO in our revised version to better highlight the differences between the two approaches.
>
> > Performance on other tasks
>
> Thank you for your question. We have focused on math reasoning tasks, which we consider to be both representative and challenging, and they sufficiently demonstrate the effectiveness of PROS. Regarding MATH500, it is much easier than AIME we have currently evaluated. In fact, the cold start checkpoint (Qwen3-8B) has already achieved over 90 on MATH500, making it less suitable for reflecting differences among methods.
>
> > Can the authors discuss potential strategies to mitigate this effect?
>
> We believe one of the most direct strategies to mitigate this effect is to forcefully introduce a small portion of the original queries into the training batch at each iteration, similar to experience replay. This could help maintain diversity and prevent over-concentration on previously reused prefixes.
>
> > Could the paper also compare simpler alternatives (e.g., variance-based scoring) to justify whether BI offers sufficient benefit to warrant its added complexity?
>
> Thank you for your suggestion. In our main paper, we already include Priority Sampling and Dynamic Sampling, which are simpler selection methods widely used in many open-source LLMs (e.g., Kimi K2 and Seed-thinking). Additionally, we have directly ablated the Bayesian Inference (BI) mechanism in the PROS-ablation experiment.
>
> As a supplement to your suggestion, we also introduced a reward variance-based selection method on top of PROS prefix reuse. This method records historical reward data and greedily selects the one with the highest reward variance. To ensure sufficient diversity, we also incorporated an ε-greedy strategy. We include the results of this experiment in the revised version (Table 5), and have excerpted the results below for your convenience:
>
> | Method                                    | AIME24 | AMC23  |
> |-------------------------------------------|--------|--------|
> | PPO                                       | 21.25  | 52.59  |
> | &nbsp;&nbsp;w/ dynamic                    | 25.73  | 61.19  |
> | &nbsp;&nbsp;w/ replay                     | 23.75  | 53.73  |
> | &nbsp;&nbsp;w/ prior                      | 24.90  | 60.44  |
> | &nbsp;&nbsp;w/ PROS (reward variance selection) | 25.72  | 61.81  |
> | &nbsp;&nbsp;w/ PROS                       | **27.40**  | **62.12**  |
>
>
> Finally, we would like to emphasize that our core contribution is the introduction of prefix reuse. BI-based selection is an additional plugin that further enhances the performance of prefix reuse.
>
> > In Figure 5(a), some baselines (e.g., Vanilla PPO) appear to have been early-stopped before convergence. Could the paper clarify the stopping criteria?
>
>
> Thank you for your question. Based on preliminary experiments and other related work, we determined that 400 training steps is sufficient for convergence for most methods. However, we acknowledge that Vanilla PPO exhibits some unusual performance fluctuations in the final steps on DAPO-train. To address your concern, we continued training from the last checkpoint, and the results are shown below:
>
> | Step  | 400    | 405    | 410    | 415    | 420    | 425    | 430    | 435    |
> |-------|--------|--------|--------|--------|--------|--------|--------|--------|
> | Performance | 72.63  | 71.87  | 72.25  | 71.49  | 72.33  | 72.17  | 70.19  | 70.65  |
>
>
> > how the baseline methods perform under the same hyperparameter variations as in Table 2 & 3
>
> Thank you for your question. We are not entirely sure we understand the question.
> Table 2 presents a comparison between our proposed methods and baselines, as shown in Table 1 (i.e. the main experiment), but with a different LLM setting. Table 3 includes ablation studies conducted on our proposed PROS, and there are no $\sigma$
> or $\lambda$ hyperparameters for the other baselines.

---

> > ### Comment · Reviewer_NFuG · 2025-11-27
> >
> > Thank you for the response, some of concerns are addressed. I will wait until the end of the discussion period to make the final decision about my score.

---

> > > ### Author Response · Authors · 2025-11-28
> > >
> > > Thank you for the update. We are pleased to know that our response has addressed some of your concerns.
> > >
> > > In order to further improve our manuscript and clear up any potential misunderstandings, could you please kindly specify which concerns still remain? Knowing this would be very helpful for us to better polish our work and address the specific weaknesses you perceive.
> > >
> > > Thanks again for your help in the review process.

---

### Official Review · Reviewer_NRn5 · 2025-10-31

**Soundness:** 3
**Presentation:** 3
**Contribution:** 3
**Rating:** 4
**Confidence:** 3

**Summary:**

The manuscript proposes to improve the generation phase of Reinforcement Learning by not generating each complete trajectory separately but instead identifies promising prefixes for a given problem and then only generates additional trajectories based on those prefixes. The author call their approach PROS (Prefix Reuse for On-policy Sampling) and the combination of problem and trajectory prefix Augmented Queries. These new queries can then be added to the training dataset in subsequent training steps and can be treated like regular problems, however with less redundant computation necessary.

Prefixes are chosen based on uncertainty, for which token-level entropy is used, as well as value function offered by the critic model, if available in the given setup. To select Augmented Queries, PROS uses a hierarchical Bayesian model to estimate their pass rates and focuses on queries, where reward uncertainty is the highest to improve exploration. Additionally optimization techniques such as exponential forgetting of rewards are employed, so that more recent Augmented Queries are selected, giving more weight to the improved policy.

**Strengths:**

* address a relevant topic, i.e. how to improve the efficiency of the generation phase of the RL pipeline, and is also timely
* manuscript is mostly well written with the idea being clearly presented and the theoretical underpinnings derived well
* reasonable evaluation, which includes a similar reasonable ablation study

**Weaknesses:**

The evaluation is reasonable (two datasets, three baselines and a little bit of ablation, which is also reasonable), however it is also somewhat limited, since only a single, small, model is evaluated. Some additional ablation study on an ever smaller model is provided. Additionally the computational environment is not discussed even though metrics such as wall clock time are reported.
The analysis is at times spotty, i.e. not comprehensive enough, for example one of the motivations was to address redundancy in the generation (at least the last part of Section 2 as well as the first part of the related work section give that impression), but the token cost increase (at least for AIME-Old).
Some notation issues, such as using certain symbols twice or inconsistent parameter order, muddy the otherwise clear writing.

minor issues:
* introduction, page 1:
  * first paragraph: reference for CoT is missing
  * second paragraph: "PPO(Schulman" and "GRPO(Shao" - missing whitespace
* chapter 2:
  * page 3, first paragraph: "And the objective" - language, a sentence should not start with "And"
  * page 3, Redundancy in Early Reasoning Steps: "on DAPO-Train dataset" - language: add a "the" before DAPO-Train?
  * Figure 2b/c: It might be worth discussing the two metrics a little bit, since they seem to go into different directions - normalized edit distance: smaller values indicate higher similarities, where it is the opposite for ROUGE-L?
    * also ROUGE-L (text) is written inconsistently (vs. Rouge-L in Figure 2c)
* 3.1, last paragraph: "remains the on-policy nature of RLVR algorithm" - language: sounds strange to me, maybe "retains the on-policy nature of the RLVR algorithm"
* Figure 3:
  * You might want to careful with the notation. In Section 2 $y_i$ is an individual thought whereas here it represents a whole trajectory/rollout.
  * "Promising prefix identification" - the other titles of the rounded rectangles are capitalized
* 3.2, page 5, first paragraph: "leverage the readily available value function learned by critic model for those actor-critic methods"
  * you claim that PROS can be easily be plugged into GRPO, but GRPO does not have a critic models, but derives its value function differently: in the subsequent paragraph this is presented more clearly
* 3.2, last paragraph:
  * "with highest token-level" - language: add "the" before "highest"?
  * maybe explain what $H$ is; here it is probably the log-likelihood, but later it is used for the history (in 4.2)
* Figure 4: What is $\mathbb{S}_j$? Probably the children of parent j. (After reading further it is discussed in 4.2, so after the figure is displayed.
* 4.2:
  * Proposition 4.1: "V)where" in equation, at least a whitespace is missing, maybe even an additional comma
  * sometimes the History $H$ is $(n, s)$ (at the beginning, Algorithm 1) and sometimes the other way around (for the propositions); similar issues in Appendix B.2
* 5.1: title is probably not necessary, if Section 5 has not other subsections
* references:
  * cited differently than other arXiv references (can only be surmised from the URL); additionally consider capitalizing the titles to be consistent, especially abbreviations and proper names:
    * [Dumitrascu et al. 2018]
    * [Fedus et al. 2020]
    * [He at al. 2025]
    * [Hong et al. 2022a/b]
    * [Lambert et al. 2025]
    * [Liang et al. 2021]
    * [Liu et al. 2025a/b]
    * [Narvekar et al. 2020]
    * [OpenAI et al. 2024]
    * [Polson et al. 2013]
    * [Razin et al. 2025]
    * [Schaul et al. 2016]
    * [Schulman et al. 2017]
    * [Shao et al. 2024]
    * [Snell et al. 2024]
    * [Soviany et al. 2022]
    * [Wang et al. 2021]
    * [Wang et al. 2017]
    * [Xi et al. 2024]
    * [Yu et al. 2025]
  * consider capitalizing the titles to be consistent, especially abbreviations and proper names:
    * [V Team et al. 2025b]
* B.3:
  * title: It is probably Proposition 4.2 instead of 4.1
  * page 16:
    * line 810: unnecessary whitespace at the beginning of the line
    * equation cuts into the margin
    * line 826: comma should probably be at the end of the previous line, after the equation
* C:
  * uses PROST instead PROS
  * line 843: "from replay buffer" - language: add a "the" before "replay"; same line: "by current policy" - add a "the" before "current"

**Questions:**

* Figure 2a: I did not understand what fraction is shown in that subfigure. Of the whole policy optimization algorithm?
* What is the computational overhead for the selection and reward estimation?

---

> ### Author Response · Authors · 2025-11-15
>
> First, we would like to express our sincere gratitude for your careful evaluation of our work. We deeply appreciate the insightful comments and suggestions provided. In response, we want to clarify the key concerns you raised.
>
> > the evaluation is reasonable, but is also somewhat limited, since only a single, small, model is evaluated.
>
> Thank you for your feedback. As you pointed out, in our main experiments, we considered two datasets, two RLVR methods, and three baselines + one ablation, which together form 16 experimental settings (2 * 2 * 4). Each experiment was trained for 400 steps, with each costing over $20,000 based on equivalent compute resources from Google Cloud. We believe this ensures the comprehensiveness and rigor of our evaluation. Additionally, we have included another model to validate generalization.
>
> > the computational environment is not discussed
>
> Thank you for your comment. As mentioned in Appendix C (line 848), all experiments were conducted in an identical computational environment, ensuring that all engineering parameters were kept consistent across all runs.
>
> > the token cost increase for AIME-Old
>
> As explained in line 411, the increase in token cost for AIME-Old is primarily due to the fact that vanilla PPO did not achieve ideal length scaling on this dataset. A key goal of RLVR training is length scaling, where the policy learns to generate longer CoT tokens for effective test-time scaling. However, as shown in Figure 6, the average length for vanilla PPO decreases as training progresses, causing its performance to plateau. In contrast, our method continuously increases the number of tokens generated per iteration, which aligns with our expectations for improved scaling.
>
> Regarding the statement that PROS reduces redundancy, this is indeed implicit in the design of our method, where we reuse prefixes from previous rollouts. In this regard, PROS shares similarities with experience replay, though it focuses on reusing prefixes instead of entire rollouts.
>
> > Formatting and Grammatical Errors
>
> Thank you for such detailed and thorough review! We greatly appreciate the time you took to identify various formatting and grammatical errors. We have carefully addressed all the issues you raised, and these corrections have been incorporated into the revised version of the paper.
>
> > What is the computational overhead for the selection and reward estimation?
>
> Thank you for your question. The computational overhead for the selection and reward estimation is negligible compared to the RLVR algorithm. Since our training framework is based on Ray, and both selection and reward estimation are purely CPU operations, we leverage Ray's parallelism to hide the time cost within the actor and critic updates. In short, there is no additional time overhead.

---

### Official Review · Reviewer_yLPZ · 2025-11-03

**Soundness:** 4
**Presentation:** 3
**Contribution:** 3
**Rating:** 8
**Confidence:** 3

**Summary:**

The paper tackles a bottleneck in RLVR: the high cost of on-policy rollouts. The authors observe that many rollouts for the same prompt share identical or near-identical prefixes. PROs leverages this redundancy by reusing verified prefixes from previous rollouts constructing augmented queries that begin from these prefixes, and then sampling only the continuation (suffix). This idea is known before. See below.
The authors talk about a hierarchical Bayesian selection scheme to identify which prefixes are most promising to reuse, prioritizing uncertain or under-explored augmented queries. That was interesting and to my knowledge novel.

 Comparison:

SPEC-RL (Liu et al., 2025) also reuses verified rollouts but does so at a coarser granularity, caching *entire* rollouts or long segments. PROS’s is finer-grained.
 AR3PO (Zhang et al., 2025) proposes *response reuse* (retraining on previously verified answers) but does not address redundancy within rollouts themselves. PROS directly targets intra-trajectory redundancy.
TreePO as the name suggests (MAP/ByteDance, 2025) exploits shared prefixes structurally through tree-based rollout planning.

There are earlier works going to Kearns et al I believe. But overall,  PROS represents a clear step beyond existing efficiency methods by focusing on *within-trajectory reuse*, introducing a principled selection mechanism, and maintaining on-policy validity.

Despite minor concerns about exploration and theoretical framing, the contribution is substantial, the experiments convincing, and the writing clear. PROS is likely to influence future work on efficient LLM-based reinforcement learning.

**Strengths:**

Clear motivation and empirical grounding: The observation that many rollouts share prefixes is both intuitive and empirically demonstrated.
Genearlity: The prefix-reuse mechanism is orthogonal to the underlying policy architecture or reward type.
Soundness:  The Bayesian selection framework is well justified and avoids trivial over-reuse of easy prefixes.
Impact: Implementation is simple and yields immediate compute savings in RLVR systems—highly relevant for scaling large-model training.

-

**Weaknesses:**

Exploration bias By reusing existing prefixes, PROS may over-concentrate training around known reasoning paths, potentially limiting discovery of novel strategies. While the Bayesian selector mitigates this, empirical diversity metrics would strengthen the case.

Minimal theoretical framing: The paper remains largely empirical; a more formal analysis of sample complexity improvement or convergence would enhance rigor.

**Questions:**

none

---

> ### Author Response · Authors · 2025-11-15
> **Response by Author**
>
> Thank you for your positive and encouraging feedback! We greatly appreciate the time and effort you put into reviewing our paper and for recognizing the contributions of our work. Your thoughtful comments and the high rating have been a great motivation for us. In the following, we address some questions you’ve raised.
>
> > PROS may over-concentrate training around known reasoning paths
>
> We acknowledge that exploration bias may arise from reusing existing prefixes.
> However, most current work in LRMs which aimed at promoting diversity ultimately focuses on improving end-to-end performance on the test set. Therefore, we prioritize improving test set performance as the most reliable indicator of our method's effectiveness.
> In future work, we will explore incorporating diversity-related regularization methods (e.g., randomly dropping existing prefixes) to further address this issue.
>
> > a more formal analysis of sample complexity improvement or convergence would enhance rigor.
>
> Thank you for your suggestion. However, given the complexity of the RLVR framework and the presence of large language models (LLMs), it is very difficult to provide a formal convergence proof. We would like to leave it for the future work

---

### Official Review · Reviewer_cmZx · 2025-11-07

**Soundness:** 3
**Presentation:** 3
**Contribution:** 3
**Rating:** 6
**Confidence:** 3

**Summary:**

The paper targets the **on-policy rollout bottleneck** in RL with verifiable rewards (RLVR) for reasoning: many on-policy rollouts for the same query share long identical prefixes, so generation time is wasted regenerating early steps. The proposed method, **PROS**, (1) identifies “promising” prefixes in past *correct* rollouts using readily available signals (token-level entropy and critic value), appends them to the original query to create **Augmented Queries (AQs)**, and then trains *only on the newly generated continuation*—preserving on-policy updates while saving generation compute; and (2) runs a **hierarchical Bayesian selector** over the growing parent→children (query→AQ) tree to prefer items with uncertain pass rates (≈0.5), sharing strength between a parent query and its AQs.
PROS integrates as a plugin to PPO/GRPO and is evaluated primarily on math benchmarks (AIME24, AMC23), reporting Pass@1 and compute trade-offs; Dynamic Sampling, Prioritized Sampling, and Experience Replay serve as baselines.

LLM usage disclosure: I used a large language model to reorganize wording and improve clarity in this review. The judgments expressed are my own.

**Strengths:**

* **Originality/Significance:** The paper addresses a critical RLVR bottleneck. Its novelty lies in integrating prefix reuse (which *creates* a structured dataset) with a Bayesian selector *tailored* to that hierarchical structure.
* **Methodology:** The method is sound. Applying gradients only to the new continuation preserves the on-policy nature of RLVR. The hierarchical Bayesian selector with exponential forgetting is a principled way to share statistics and adapt to the non-stationary policy.
* **Empirical Quality:** Experiments use strong, relevant baselines like Dynamic Sampling. Figure 5 shows PROS achieves a better performance-cost trade-off, outperforming Dynamic Sampling while being 2-4x faster in wall-clock time. An ablation confirms the selector's value over random sampling.
* **Clarity:** The paper is well-written, using Figures 1 and 3 to effectively illustrate the core concepts of prefix redundancy and the PROS pipeline.

**Weaknesses:**

* **Limited Evaluation Metric (Pass@1):** The evaluation relies exclusively on Pass@1. This metric fails to show if the model is learning *diverse* solutions. Pass@K (K=32/128, etc.) is a feasible and important metric that is missing.* **Unclear Final Capability vs. Efficiency:** The paper's central claim is a better performance-cost trade-off, and it excels at showing faster initial learning (Figure 5). However, the core methodological trade-off against baselines like Dynamic Sampling (DS) is not fully resolved.
* **Risk of "Hard-Tail" Starvation:** PROS explicitly targets queries with high reward uncertainty (pass rate $\approx 0.5$). This is a known efficiency strategy, but it risks starving the model of exposure to the hardest problems (pass rate $\approx 0$). In contrast, DS is *specifically designed* to focus on this hard-tail. The experiments, which are limited in training time, do not establish whether PROS's efficiency-first approach can match the final performance ceiling of a capability-first method like DS, given a matched (and sufficient) compute budget. The paper is missing a clear convergence-point comparison.
* **Un-ablated Prefix Heuristics:** The prefix identification rule (entropy + value) is purely heuristic . The paper provides **no ablation** to justify this choice over simpler ones (e.g., value-only). The main "PROS-ablation" only tests the *selector* (Component 2), not the *prefix identification* (Component 1).

**Questions:**

1. **Pass@k and budget matching.** Please report **Pass@k** (e.g., k=8/32/128) in addition to Pass@1, with matched budgets, to separate exploration breadth from sampling reweighting gains. (Current results focus on Pass@1. )

2. **Convergence & final comparison.** Provide **longer-horizon** training curves and **final** metrics comparing PROS to **Dynamic Sampling** and other baselines under **identical compute budgets**. Check if methods like Dynamic Sampling eventually catches up or surpasses PROS (as hinted on some setups) and discuss whether PROS’s early gains diminish over time.

3. **Hard-subset analysis.** Report performance on the **hardest part** of items and discuss whether mid-difficulty targeting (pass rate ≈0.5) reduces exposure to very hard queries (pass rate ≈0). (Selector targets ≈0.5 region. )

4. **Prefix heuristics ablation.** Ablate the **prefix-picking** rule: (a) value-only, (b) entropy-only, (c) random correct-prefix; include efficiency and accuracy. (Heuristic defined in Sec. 3.2. )

5.  **Generality to Other Domains:** Do you have any results applying PROS to a non-mathematical RLVR task, such as code generation?

---

> ### Author Response · Authors · 2025-11-15
> **Response by Author (Part 1)**
>
> We would like to thank you for the thoughtful and constructive feedback. We appreciate the time and effort you put into reviewing our work. In the following, we address the questions raised and provide clarifications and additional details to resolve any concerns.
>
>
> > Limited Evaluation Metric (Pass@1) & Risk of "Hard-Tail" Starvation
>
> We appreciate the suggestion of including additional evaluation metrics. In response, we have calculated Pass@32 results, which are now included in the revised version of the paper (Table 4). For your convenience, we provide the summarized results below:
>
> | Method                           | AIME24 | AMC23 | #Tokens (M) | #Time (min) |
> |-----------------------------------|------------|-------------|-------------|-------------|
> | PPO                               | 57.67      | 90.03       | 10.24       | 9.17        |
> | w/ Dynamic Sampling               | 63.89      | 90.69       | 11.07       | 17.62       |
> | w/ Experience Replay              | 63.15      | 90.05       | 8.93        | 8.98        |
> | w/ Priority Sampling              | 64.55      | 91.41       | 11.35       | 9.76        |
> | w/ PROS-ablation                  | 62.64      | **92.13**   | 7.92        | 9.29        |
> | w/ PROS                           | **66.05**  | 90.87       | 8.49        | 9.77        |
> |-----------------------------------|------------|-------------|-------------|-------------|
> | GRPO                              | 59.14      | 90.62       | 9.27        | 6.58        |
> | w/ Dynamic Sampling               | 63.82      | 91.09   | 11.16       | 19.45       |
> | w/ Experience Replay              | 63.48      | 91.05       | 7.60        | 6.88        |
> | w/ Priority Sampling              | 64.77  | 91.02       | 9.55        | 6.87        |
> | w/ PROS-ablation                  | **65.44**  | **91.24**   | 7.87        | 7.00        |
> | w/ PROS                           | 65.01  | 90.54       | 8.46        | 7.57        |
>
> Based on the results presented above, we believe that the risk of "hard-tail" starvation does not significantly affect our method.
>
> Regarding the point that "DS is specifically designed to focus on the hard-tail", we would like to kindly clarify that Dynamic Sampling does not specifically target the hard-tail. You may refer to Priority Sampling, which is more directly aimed at the hard-tail. As seen in the results, while Priority Sampling does show strong performance in Pass@K, it still does not outperform PROS in terms of efficiency and effectiveness.
>
>
> > Unclear Final Capability vs. Efficiency
>
> We want to first emphasize that our proposed method is fundamentally designed to improve efficiency, which is the key motivation behind the "prefix reuse" approach. Regarding final capability, as shown in Table 1, when trained for the same number of steps (400 steps for each method), PROS demonstrates more stable performance than Dynamic Sampling.
>
> We have observed that for most methods, 400 steps is nearly sufficient for convergence, as performance on the test set shows no significant improvement during the last 50 steps, instead exhibiting small oscillations or even a decline. We acknowldge that we cannot prove the final convergence without empirically training for more iterations (maybe about 2000steps), which would be too expensive to afford. Each experiment in our study (i.e. each line in Figure 5) cost approximately $20,000 (Based on equivalent compute resources from Google Cloud (https://cloud.google.com/products/calculator)). For the Dynamic Sampling method, the training cost exceeded $40,000. We believe this experimental scale aligns well with the computational budgets used in other comparable studies [1] [2].
>
> [1] ReMax: A Simple, Effective, and Efficient Reinforcement Learning Method for Aligning Large Language Models (ICML)
> [2] Back to Basics: Revisiting REINFORCE Style Optimization for Learning from Human Feedback in LLMs (ACL)

---

> ### Author Response · Authors · 2025-11-15
> **Response by Author (Part 2)**
>
> > Un-ablated Prefix Heuristics：
>
> In fact, the choice of the prefix identification rule (entropy + value) was made somewhat ad-hoc, without extensive hyperparameter tuning. Our main idea is prefix reuse, and as shown in the main experiments (Table 1), even with current simple heuristic, we achieved strong performance, demonstrating the effectiveness of prefix reuse itself.
>
> We appreciate the suggestion and conduct two additional ablation experiments focusing on the prefix selection mechanism (Component 1) using the Qwen3-4B model. The results are presented in Appendix (Table 4). We also provide them here for your convenience.
>
> | Method                                    | AIME24 | AMC23  |
> |-------------------------------------------|--------|--------|
> | PPO                                       | 21.25  | 52.59  |
> | &nbsp;&nbsp;w/ PROS (value-based prefix identification)                       | 24.68 | **63.79** |
> | &nbsp;&nbsp;w/ PROS (entropy-based prefix identification)                      | 25.72 | 61.81  |
> | &nbsp;&nbsp;w/ PROS                       | **27.40**  | 62.12  |
>
> The entropy-based prefix identification selects the position with highest token-level entropy (i.e. the same selection heuristic for GPRO w/ PROS). The value-based prefix identification selects the position with highest value given by critic. The results show that all variants outperforms the baseline by a large margin.
>
>
> > discuss whether mid-difficulty targeting (pass rate ≈0.5) reduces exposure to very hard queries (pass rate ≈0).
>
> We appreciate your question. Indeed, the purpose of the selector in our method is to increase the exposure to medium-difficulty problems (with pass rate ≈ 0.5) while reducing the occurrence of very easy and very hard problems. The advantages of this approach have already been discussed in [3], where it was shown that focusing on medium-difficulty queries improves learning efficiency and overall performance.
>
>
> [3] Online Difficulty Filtering for Reasoning Oriented Reinforcement Learning
>
>
> > Generality to Other Domains
>
> Currently, we have focused on math reasoning tasks, which we consider to be a representative and challenging domain that sufficiently demonstrates the effectiveness of PROS.
> In addition, we conducted a small-scale experiment on code generation (1.5B) in our preliminary experiment, where prefix reuse mechanism similarly led to stable performance improvements. In the future, with more time and computational resources, we would expand our experiments to other domains for further validation.

---

> ### Comment · Reviewer_cmZx · 2025-11-26
> **Response by Reviewer cmZx**
>
> My concerns are partially solved, but it seems that no revision has been made to the paper by now. (At least, Table 4 is missing. ) Would you please double check if you have uploaded the revised version?

---

> > ### Author Response · Authors · 2025-11-27
> > **New Revision Uploaded**
> >
> > We apologize that the previous attempt to upload the PDF might have failed due to network issue. We have now re-submitted the revised version.

---

### Author Response · Authors · 2025-11-15
**Clarification on Revised Version (Nov 15)**

We have made several modifications in the revised version of the paper:

1. Fixed all formatting errors identified by Reviewer NRn5.

2. Added two additional experimental results in Appendix D in response to reviewers NFuG and cmZx.

---

### Author Response · Authors · 2025-11-27

Dear Reviewers,

Thank you very much for the time and effort you have invested in reviewing our submission. We sincerely appreciate your thoughtful feedback, which have been extremely helpful to improve the paper.

We would like to kindly remind that the discussion period will end in 4 days. If there are any remaining questions, clarifications, or points that would benefit from further explanation or discussion, please feel free to let us know, we are very happy to respond promptly.

Thank you again for your contributions and for considering our work.

Best regards

---

### Meta-Review · Area_Chair_o8sQ · 2025-12-16

**Summary:**

The paper proposes PROS, a plug-in paradigm for RLVR that reuses verified rollout prefixes to form augmented queries and trains only on continuations, paired with a hierarchical Bayesian selector targeting mid-difficulty items (pass rate ≈0.5). Reviewers generally find the motivation strong, the selector principled, and the empirical gains convincing on math reasoning tasks. The main weaknesses raised were: limited evaluation scope (primarily AIME24/AMC23, Qwen3-8B and a smaller 4B model), concerns about exploration bias and potential hard-tail starvation, limited analysis of final capability under longer horizons, and numerous presentation/notation issues. I read the rebuttal and revised manuscript; the authors added Pass@32 results (Appendix D, Table 4), ablated prefix identification heuristics and a simpler reward-variance selector (Table 5), clarified early stopping and length scaling behavior, and stated computational environment and overhead details (Appendix C). These additions substantively address exploration/diversity and selection-complexity concerns and resolve most presentation and methodological questions. Remaining limitations are the domain scope (math-only), absence of longer-horizon convergence comparisons under equal compute, and moderate novelty relative to related/concurrent work (e.g., response/tree-based reuse). Given the strong performance–cost trade-offs (Figure 5; Tables 1–2) and added ablations, I recommend accept (poster). I would like to recommend spotlight, but no spotlight this year.

**Reviewer Concerns:**

### Reviewer_NRn5
- **Concern**:
  Evaluation limited (single small model, scope), computational environment not discussed, and unclear token cost increase; numerous notation/formatting issues; overhead of Bayesian selection unknown.

- **Why Unresolved**:
  Partially resolved. The authors added a second (smaller) model (**Qwen3-4B**) and clarified environment in **Appendix C**; they explained token-cost/length scaling and reported negligible overhead due to Ray-based CPU operations. However, evaluation remains math-centric and lacks broader domains (e.g., code) and longer-horizon convergence under matched compute.

- **Impact on Decision**:
  **Moderate**. With environment clarified and overhead negligible, the main remaining limitation is scope and long-horizon capability. This precludes a spotlight but does not justify rejection given practical gains and improved clarity.

---

### Reviewer_NFuG
- **Concern**:
  Limited datasets (no code/MATH500), potential diversity reduction, need for simpler selection baselines, early-stopping clarity, and fairness in hyperparameter studies; possible overlap with TreeRPO.

- **Why Unresolved**:
  Mostly resolved. The authors added a reward-variance selector baseline (**Table 5**), clarified stopping by extending PPO beyond 400 steps, and discussed diversity mitigation strategies. They argued MATH500 is too easy for discrimination and noted TreeRPO as concurrent and methodologically distinct. The remaining limitation is lack of non-math domains beyond a brief 1.5B code pilot not fully reported.

- **Impact on Decision**:
  **Low to moderate**. Adequate ablations and clarifications support acceptance; limited cross-domain validation keeps it at poster level.

---

### Reviewer_cmZx
- **Concern**:
  Missing Pass@K, risk of hard-tail starvation due to targeting ≈0.5 pass rate, unclear final capability vs. efficiency, and no ablation of prefix-picking heuristics; generality to other domains.

- **Why Unresolved**:
  Substantially resolved. Pass@32 added (**Table 4**) and shows PROS competitive/strong while maintaining efficiency; prefix-picking ablated (**Table 5**). The authors’ efficiency-first claim is credible, but longer-horizon matched compute comparisons remain absent. Domain generality is still mainly anecdotal.

- **Impact on Decision**:
  **Low**. The added metrics and ablations overcome core objections; lack of long-run comparisons mainly affects ranking (poster vs. spotlight).

---

### Reviewer_yLPZ
- **Concern**:
  Exploration bias and limited theoretical framing.

- **Why Unresolved**:
  Partially resolved. Empirical Pass@32 and diversity discussions help; a formal convergence or sample complexity analysis remains out of scope, as acknowledged.

- **Impact on Decision**:
  **Low**. The empirical contributions and practical relevance outweigh the lack of theory for acceptance.

**Reviewer Scores:**

Reviewer_NRn5
- Original Score: 4
- Expected Score After Discussion: 4 or 6
- Rationale: The rebuttal corrected presentation issues, clarified environment and overhead, explained length scaling, and added ablations. Scope remains narrow (math-only) and no long-horizon convergence under matched compute; thus a modest bump but still below a clear accept.

---

Reviewer_NFuG
- Original Score: 6
- Expected Score After Discussion: 6
- Rationale: Concerns on selection complexity and early stopping were addressed (variance-based baseline; extended training steps), and diversity mitigation discussed. Lack of multi-domain experiments keeps the score at solid borderline-accept.

---

Reviewer_cmZx
- Original Score: 6
- Expected Score After Discussion: 6
- Rationale: Pass@32 and prefix-heuristic ablations directly addressed the key objections; efficiency claims remain strong. Missing longer-horizon matched compute prevents a higher score.

---

Reviewer_yLPZ
- Original Score: 8
- Expected Score After Discussion: 8
- Rationale: Positive assessment stands; diversity addressed empirically, but theoretical analysis remains limited as acknowledged.

---

### Decision · Program_Chairs · 2026-01-26

Accept (Poster)